# Biomarkers of mortality in adults and adolescents with advanced HIV in sub-Saharan Africa

Victor Riitho[1,2], Roisin Connon [3], Agnes Gwela[4], Josephine Namusanje[5], Ruth Nhema[6], Abraham Siika[7], Mutsa Bwakura-Dangarembizi [6], Victor Musiime [5,8], James A. Berkley [4], Alex J. Szubert [3], Diana M. Gibb[3], A. Sarah Walker[3], Nigel Klein [6] & Andrew J. Prendergast [1] ✉

One-third of people with HIV in sub-Saharan Africa start antiretroviral therapy (ART) with advanced disease. We investigated associations between immune biomarkers and mortality in participants with advanced HIV randomised to cotrimoxazole or enhanced antimicrobial prophylaxis in the Reduction of Early Mortality in HIV-Infected Adults and Children Starting Antiretroviral Therapy (REALITY) trial (ISRCTN43622374). Biomarkers were assayed using ELISA and Luminex. Associations between baseline values and all-cause 24-week mortality were analysed using Cox models, and for cause-specific mortality used Fine & Gray models, including prophylaxis randomisation, viral load, CD4, WHO stage, age, BMI, and site as covariates; and weighted according to inverse probability of selection into the substudy. Higher baseline CRP, IFN-γ, IL-6 and IP-10 were associated with higher all-cause mortality; and higher IL-23, IL-2 and RANTES with lower all-cause mortality. Associations varied by cause of death: tuberculosis-associated mortality was most strongly associated with higher CRP and sST2, and cryptococcosis-associated mortality with higher IL-4 and lower IL-8. Changes in I-FABP ($p = 0.002$), faecal alpha-1 antitrypsin ($p = 0.01$) and faecal myeloperoxidase ($p = 0.005$) between baseline and 4 weeks post-ART were greater in those receiving enhanced versus cotrimoxazole prophylaxis. Our findings highlight how the immune milieu shapes outcomes following ART initiation, and how adjunctive antimicrobials can modulate the gut environment in advanced HIV.

Antiretroviral therapy (ART) has greatly decreased the burden of HIV/AIDS-related morbidity and mortality in sub-Saharan Africa[1,2]. According to UNAIDS estimates, of 20.7 million people living with HIV in Eastern and Central Africa by the end of 2020, 87% were aware of their HIV status, 83% had access to ART, and about 90% of those achieved viral suppression[3]. Whilst increased availability of testing services and WHO "treat all" guidelines have encouraged early ART initiation, one-third of people still start treatment with advanced HIV, as defined by a CD4 T-cell count <200 cells/mm³ or a WHO clinical stage 3 or 4 event[4]. Early mortality is high in this group, with 8–26% of

[1]Blizard Institute, Queen Mary University of London, London, UK. [2]Center for Epidemiological Modelling and Analysis (CEMA), Institute of Tropical and Infectious Diseases (UNITID), University of Nairobi, Nairobi, Kenya. [3]MRC Clinical Trials Unit at UCL, London, UK. [4]KEMRI-Wellcome Trust Research Programme, Kilifi, Kenya. [5]Joint Clinical Research Centre, Kampala, Uganda. [6]University of Zimbabwe, Harare, Zimbabwe. [7]Moi University, Eldoret, Kenya. [8]Department of Paediatrics and Child Health, Makerere University, Kampala, Uganda. ✉e-mail: a.prendergast@qmul.ac.uk

those initiating ART with advanced HIV dying within the first three months of treatment[5,6].

Early deaths are predominantly due to infections as a consequence of multiple underlying pathological derangements, including immune dysfunction; chronic inflammation; immune reconstitution inflammatory syndrome (IRIS), leading to unmasking of opportunistic co-infections; malnutrition; and HIV enteropathy, which enables translocation of gut lumen microbes to the systemic circulation[7,8]. Biomarkers of inflammation are independently associated with mortality in HIV infection, even after initiation of ART[9,10]. However, few studies have focused on advanced HIV in sub-Saharan Africa, where the complex interplay between HIV replication, co-infections, enteropathy and immune dysfunction may perturb inflammatory pathways more profoundly; furthermore, few studies have explored associations between inflammatory biomarkers and cause-specific mortality[11].

We previously showed in the REALITY trial, conducted among adults, older children and adolescents initiating ART in sub-Saharan Africa with advanced HIV (CD4 < 100 cells/mm³), that an enhanced package of antimicrobial prophylaxis reduces mortality by 27% over the first 24 weeks on ART, by reducing tuberculosis, cryptococcosis, candidiasis and deaths from unknown causes[12]. However, we lack understanding of whether, and how, this antimicrobial package might also confer benefits by targeting underlying inflammation and enteropathy.

Here, we investigate the effects of baseline inflammation, immunoregulation and enteropathy on mortality in the REALITY trial, and investigate whether enhanced infection prophylaxis modulates these pathways. Our hypotheses were that i) biomarkers of inflammation, immunoregulation and enteropathy are independently associated with mortality; ii) specific baseline biomarker signatures distinguish different causes of death; and iii) enhanced antimicrobial prophylaxis alters biomarkers of inflammation and enteropathy.

## Results

Biomarkers were measured in 599 participants with advanced HIV (CD4 < 100 cells/mm³) enroled in the REALITY trial, of whom 169 died by 24 weeks (median 6 (IQR 3–10) weeks to death), and 430 survived (case-cohort design; Supplementary Fig. 1). The baseline characteristics of participants and their biomarker concentrations prior to ART initiation are shown in Table 1. Those who died, compared to those who survived, were significantly older and more wasted, with a lower CD4 count and higher WHO disease stage; mortality also differed by centre. Sub-study participants randomised to enhanced prophylaxis (cotrimoxazole plus isoniazid/pyridoxine, azithromycin, albendazole, and fluconazole) versus standard-of-care (cotrimoxazole prophylaxis) had lower mortality (HR 0.55, 95% CI 0.37, 0.81), as previously reported for the whole trial[12]. Of the 169 sub-study participants who died by week 24, independently adjudicated causes of death (which could be multiple) were 61 tuberculosis, 14 cryptococcosis, 21 serious bacterial infections, 53 other causes, and 70 unknown causes (Supplementary Table 1).

### Baseline inflammatory and immunoregulatory biomarkers are associated with early mortality

At ART initiation, participants who died compared to those who survived had significantly higher plasma C-reactive protein (CRP), soluble CD14 (sCD14), interferon gamma (IFN-γ), IL-18, IL-1RA, soluble suppression of tumorigenesis 2 (sST2), lipopolysaccharide binding protein (LBP) and RANTES, higher faecal myeloperoxidase (MPO), and lower plasma intestinal fatty acid binding protein (I-FABP) (Table 1). Correlations between biomarkers are shown in Supplementary Fig. 2. We used Cox regression models with backwards elimination (exit $p = 0.1$; exploratory analysis) to estimate the independent effect of each baseline biomarker on all-cause mortality, adjusting for baseline viraemia, CD4, WHO stage, age, body mass index, centre, and randomised prophylaxis. Higher CRP (adjusted HR 1.98 (95% CI 1.51–2.59) per $\log_{10}$ higher), IFN-γ (3.09, 1.55–6.16 per $\log_{10}$ higher), and interferon

gamma-induced protein 10 (IP-10) (2.29, 1.39–3.75 per $\log_{10}$ higher) at baseline, independently increased all-cause mortality; higher IL-23 (0.50, 0.32–0.80 per $\log_{10}$ higher), and RANTES (0.32, 0.17–0.60 per $\log_{10}$ higher) independently decreased mortality (Table 2). There was weaker evidence that higher IL-6 (adjusted HR 2.84, 1.00–8.06 per $\log_{10}$ higher) and lower IL-2 (0.20, 0.06–0.67 per $\log_{10}$ higher) increased mortality ($p < 0.05$ but not meeting Benjamini–Hochberg threshold accounting for multiple testing). There was no evidence that the association between biomarker levels and mortality was modified by the enhanced prophylaxis intervention (interaction $p > 0.12$). Taken together, we observed a clear pattern of biomarker associations, where higher levels of inflammatory biomarkers were independently associated with increased mortality by 24 weeks, whilst higher levels of immunoregulatory cytokines and chemokines were associated with lower risk of mortality.

### Distinct baseline biomarkers are associated with cause-specific mortality in advanced HIV

We next evaluated associations between individual biomarkers at ART initiation and cause-specific mortality, hypothesising that the associated inflammatory and immunoregulatory markers may differ by cause of death, and noting that with smaller number of events (deaths from specific causes) power is lower (Fig. 1). Higher CRP (sub-hazard ratio 2.01, 95% CI 1.20–3.39 per $\log_{10}$ higher) and sST2 (1.59, 1.09–2.31 per $\log_2$ higher) were associated with TB-associated deaths (occurring median 4 (IQR 2–9) weeks after ART initiation), whereas higher IL-4 (SHR 8.28, 1.90–36.06 per $\log_{10}$ higher) and lower IL-8 (0.23, 0.06–0.95 per $\log_{10}$ higher) were associated with cryptococcosis-associated deaths (occurring median 6 (IQR 3–9) weeks post-ART initiation). Deaths from serious bacterial infections (SBI) (median 2 (IQR 2–7) weeks post-ART initiation) were associated with higher CRP (2.20, 1.10–4.37 per $\log_{10}$ higher) and lower sCD163 (0.79, 0.63–0.98 per $\log_2$ higher). Higher IFN-γ (5.90, 1.97–17.66 per $\log_{10}$ higher) and sCD14 (1.98, 1.17–3.34 per $\log_2$ higher) and lower IL-9 (0.29, 0.16–0.52 per $\log_{10}$ higher) were associated with increased risk of death from 'other' causes (median 6 (IQR 3–10) weeks post-ART initiation); and higher IL-18 (4.51, 1.57–12.97 per $\log_{10}$ higher) and sCD14 (1.81, 1.22–2.70 per $\log_2$ higher) and lower TNFα (0.32, 0.12–0.90 per $\log_{10}$ higher), I-FABP (0.76, 0.61–0.95 per $\log_2$ higher) and RANTES (0.42, 0.20–0.86 per $\log_{10}$ higher) with deaths from unknown cause (median 6 (IQR 3–11) weeks post-ART initiation).

Having identified these individual biomarkers most strongly associated with specific causes of death, which overlapped to a large degree with the baseline biomarkers most strongly associated with all-cause mortality, we next considered whether there were other biomarker combinations that would be similarly associated with mortality (Supplementary Table 2). Best subsets regression showed that the models selected using backwards elimination for deaths from TB, cryptococcosis and severe bacterial infection had the best fit (lowest Akaike Information Criterion, AIC) of all models with the same number of independent variables. For all-cause mortality and deaths from 'other' causes, the best fitting model was the same as that selected by backwards elimination except with IL-18 substituted for IFN-γ, these being strongly correlated (Spearman rho=0.97, Supplementary Fig. 2). For all-cause mortality the difference in fit was small ($\Delta$AIC = 1.7) while for deaths from other causes the difference was modest ($\Delta$AIC = 3.4) suggesting these selected models are well supported[13]. The best model for deaths from unknown causes included IL-7 and IL-8 substituted for TNFα and IL-18, and had a larger AIC difference of 5.3, suggesting somewhat less support for the selected model.

### Clustering by baseline biomarker values identifies four distinct sub-groups

A principal components analysis including all 41 non-stool biomarkers identified 8 principal components (PCs) that together explained 78% of

**Table 1 | Baseline characteristics and comparison between those who died before 24 weeks vs remained alive at 48 weeks**

| Variable | Overall median (IQR) | Deaths before 24 weeks median (IQR) | Alive at 48 weeks median (IQR) | Hazard ratio[a] | p-value |
|---|---|---|---|---|---|
| Baseline clinical characteristics | | | | | |
| Age, years | 37 (30–43) | 38 (30–44) | 36 (30–43) | 1.12 (1.02–1.23) | 0.01 |
| Sex, female | 292 (49%) | 88 (52%) | 204 (47%) | 1.46 (1.00–2.14) | 0.05 |
| CD4 count, cells/uL | 34 (15–64) | 24 (10–46) | 38 (16–67) | 0.90 (0.84–0.97) | 0.003 |
| BMI, kg/m$^2$ [N = 593] | 19 (17–21) | 18 (16–19) | 19 (17–21) | 0.86 (0.80–0.93) | <0.001 |
| WHO Stage at enrolment: 3–4 | 351 (59%) | 130 (77%) | 221 (51%) | 2.69 (1.64–4.39) | <0.001 |
| Enhanced prophylaxis | 295 (49%) | 69 (41%) | 226 (53%) | 0.55 (0.37–0.81) | 0.002 |
| Plasma HIV RNA, log 10 copies/mL | 5.43 (5.04–5.79) | 5.55 (5.14–5.96) | 5.40 (4.97–5.71) | 1.35 (0.93–1.95) | 0.11 |
| Centre: Centre A | 222 (37%) | 39 (23%) | 183 (43%) | 1 | <0.001 |
| Centre B | 68 (11%) | 26 (15%) | 42 (10%) | 1.74 (0.95–3.19) | |
| Centre C | 29 (5%) | 13 (8%) | 16 (4%) | 1.01 (0.45–2.27) | |
| Centre D | 66 (11%) | 26 (15%) | 40 (9%) | 3.37 (1.87–6.09) | |
| Centre E | 52 (9%) | 21 (12%) | 31 (7%) | 4.58 (2.29–9.14) | |
| Centre F | 101 (17%) | 21 (12%) | 80 (19%) | 2.54 (1.32–4.89) | |
| Centre G | 61 (10%) | 23 (14%) | 38 (9%) | 2.08 (1.08–4.01) | |
| Baseline biomarkers (pg/ml unless otherwise stated) | | | | | |
| CD163 (ng/ml) [N = 597] | 838 (516–1398) | 930 (516–1507) | 812 (516–1330) | 1.01 (0.86–1.18) | 0.90 |
| CRP (mg/l) [N = 597] | 6 (2–39) | 33 (5–130) | 4 (1–20) | 2.46 (1.92–3.15) | <0.001[c] |
| sCD14 (ng/ml) [N = 597] | 3609 (2398–5120) | 4550 (3498–6247) | 3234 (2238–4765) | 2.23 (1.64–3.03) | <0.001[c] |
| I-FABP [N = 596] | 1897 (1112–3339) | 1602 (803–3311) | 1995 (1196–3340) | 0.74 (0.64–0.86) | <0.001[c] |
| IFN-γ [N = 588] | 182 (98–380) | 280 (161–632) | 154 (83–296) | 3.66 (2.06–6.50) | <0.001[c] |
| IL-2 [N = 588] | 34 (23–51) | 34 (23–51) | 34 (23–51) | 0.72 (0.32–1.61) | 0.42 |
| IL-4 [N = 588] | 44 (10–151) | 70 (10–190) | 35 (10–129) | 1.36 (0.98–1.89) | 0.07 |
| IL-8 [N = 588] | 64 (23–156) | 97 (39–233) | 54 (18–128) | 1.20 (0.86–1.67) | 0.28 |
| IL-9 [N = 588] | 147 (41–401) | 157 (33–413) | 145 (43–400) | 1.02 (0.74–1.41) | 0.91 |
| IL-18 [N = 588] | 190 (103–409) | 326 (169–673) | 160 (90–310) | 3.88 (2.20–6.84) | <0.001[c] |
| IL-23 [N = 588] | 490 (164–1274) | 424 (119–1166) | 539 (185–1292) | 0.85 (0.60–1.19) | 0.34 |
| IP-10 [N = 588] | 587 (240–1489) | 988 (373–2369) | 481 (208–1157) | 1.25 (0.89–1.76) | 0.19 |
| RANTES [N = 588] | 304 (201–876) | 320 (197–681) | 303 (201–982) | 0.52 (0.33–0.81) | 0.004[c] |
| D-dimer (ng/ml) [N = 597] | 3937 (2010–6722) | 4503 (2223–8299) | 3674 (1940–6083) | 1.21 (1.00–1.47) | 0.05 |
| Eotaxin [N = 588] | 64 (37–116) | 85 (49–164) | 58 (34–103) | 1.17 (0.56–2.44) | 0.67 |
| GM-CSF [N = 588] | 44 (20–98) | 50 (21–97) | 42 (19–98) | 1.16 (0.71–1.90) | 0.56 |
| GROA [N = 588] | 77 (33–157) | 80 (32–158) | 76 (33–156) | 0.89 (0.67–1.18) | 0.40 |
| IFNα [N = 587] | 0.5 (0.5–5.4) | 0.5 (0.5–4.7) | 0.5 (0.5–5.5) | 0.74 (0.54–1.01) | 0.06 |
| IL-1α [N = 588] | 0.71 (0.71–1.08) | 0.71 (0.71–2.27) | 0.71 (0.71–0.73) | 1.22 (0.82–1.81) | 0.34 |
| IL-1β [N = 588] | 50 (22–94) | 50 (24–109) | 47 (22–90) | 1.12 (0.75–1.67) | 0.58 |
| IL1-RA [N = 588] | 173 (34–1141) | 807 (34–3005) | 34 (34–681) | 1.64 (1.29–2.09) | <0.001[c] |
| IL-5 [N = 588] | 60 (34–106) | 61 (37–106) | 59 (32–106) | 0.98 (0.57–1.69) | 0.95 |
| IL-6 [N = 588] | 91 (47–198) | 114 (60–231) | 83 (42–183) | 1.75 (1.09–2.80) | 0.02 |
| IL-7 [N = 588] | 2.9 (1.0–7.2) | 3.1 (1.3–8.6) | 2.6 (1.0–6.5) | 0.89 (0.64–1.23) | 0.49 |
| IL-10 [N = 588] | 12 (3–29) | 14 (5–36) | 11 (3–26) | 1.26 (0.86–1.85) | 0.23 |
| IL-12p70 [N = 588] | 15 (12–31) | 17 (12–33) | 14 (12–30) | 1.67 (0.85–3.27) | 0.14 |
| IL-13 [N = 588] | 9 (5–19) | 11 (5–22) | 8 (5–18) | 1.53 (0.83–2.82) | 0.17 |
| IL-15 [N = 588] | 3 (3–21) | 3 (3–25) | 3 (3–19) | 1.04 (0.76–1.43) | 0.80 |
| IL-17A [N = 588] | 15 (8–26) | 16 (9–28) | 15 (8–25) | 1.20 (0.70–2.06) | 0.52 |
| IL-21 [N = 588] | 164 (69–329) | 172 (59–390) | 162 (70–319) | 1.12 (0.76–1.65) | 0.55 |
| IL-22 [N = 588] | 167 (72–382) | 176 (70–419) | 166 (72–375) | 1.12 (0.73–1.70) | 0.60 |
| IL-27 [N = 588] | 100 (34–268) | 108 (34–272) | 95 (34–263) | 1.14 (0.77–1.69) | 0.51 |
| IL-31 [N = 587] | 30 (7–190) | 57 (7–282) | 27 (7–175) | 1.06 (0.83–1.36) | 0.64 |
| LBP (mg/l) [N = 597] | 35 (22–61) | 46 (26–105) | 31 (20–50) | 1.82 (1.46–2.26) | <0.001[c] |
| MCP1 [N = 588] | 110 (61–208) | 129 (72–224) | 102 (57–198) | 0.85 (0.58–1.24) | 0.40 |
| MIP1α [N = 588] | 10 (5–25) | 12 (7–32) | 9 (4–23) | 1.10 (0.76–1.61) | 0.61 |
| MIP1β [N = 588] | 99 (64–179) | 108 (70–219) | 91 (60–174) | 0.89 (0.62–1.28) | 0.52 |

**Table 1 (continued) | Baseline characteristics and comparison between those who died before 24 weeks vs remained alive at 48 weeks**

| Variable | Overall median (IQR) | Deaths before 24 weeks median (IQR) | Alive at 48 weeks median (IQR) | Hazard ratio[a] | p-value |
|---|---|---|---|---|---|
| SDF1A [N = 588] | 702 (451–1241) | 703 (487–1257) | 700 (449–1241) | 0.79 (0.52–1.20) | 0.27 |
| sST2 [N = 597] | 13716 (8397–24731) | 27548 (13195–64521) | 11654 (7856–17856) | 1.63 (1.32–2.01) | <0.001[c] |
| TNFα [N = 588] | 64 (35–105) | 69 (35–112) | 62 (35–101) | 1.26 (0.68–2.33) | 0.46 |
| TNFβ [N = 588] | 10 (10–136) | 10 (10–239) | 10 (10–109) | 1.07 (0.81–1.41) | 0.64 |
| A1AT (mg/l) [N = 277][b] | 288 (143–564) | 420 (192–751) | 259 (136–518) | 1.33 (1.01–1.76) | 0.05 |
| MPO (ng/ml) [N = 277] [b] | 1811 (770–4771) | 2287 (770–7577) | 1769 (770–4212) | 1.37 (1.09–1.71) | 0.01[c] |
| NEO (nmol/L) [N = 276] [b] | 285 (82–1247) | 318 (86–1481) | 283 (81–1192) | 0.92 (0.77–1.09) | 0.31 |

The number of participants (N) with data for each biomarker is shown, and differs between biomarkers due to sample availability or technical failures.

[a]Hazard ratios show the association between each factor and mortality from models adjusted for prophylaxis randomisation, viral load, CD4, WHO stage, age and BMI at enrolment, and centre; and were weighted according to inverse probability of selection into the sub-study, following the case-cohort design.

[b]Stool (as opposed to plasma) biomarkers assessed only in centres A and B.

[c]44 biomarkers tested: naïve Bonferroni significance threshold = 0.05/44 = 0.00114; symbol indicates tests passing an ordered Benjamini–Hochberg (BH) threshold.

**Table 2 | Associations between baseline biomarkers and all-cause mortality (multivariable models)**

| Variable | Adjusted HR[a] | Fully adjusted HR[a] | p-value |
|---|---|---|---|
| CRP (log 10) | 2.46 (1.92–3.15) | 1.98 (1.51–2.59) | <0.001[b] |
| IFNγ (log 10) | 3.66 (2.06–6.50) | 3.09 (1.55–6.16) | 0.001[b] |
| IL-23 (log 10) | 0.85 (0.60–1.19) | 0.50 (0.32–0.80) | 0.004[b] |
| IL-2 (log 10) | 0.72 (0.32–1.61) | 0.20 (0.06–0.67) | 0.009 |
| IL-6 (log 10) | 1.75 (1.09–2.80) | 2.84 (1.00–8.06) | 0.0498 |
| IP-10 (log 10) | 1.25 (0.89–1.76) | 2.29 (1.39–3.75) | 0.001[b] |
| RANTES (log 10) | 0.52 (0.33–0.81) | 0.32 (0.17–0.60) | <0.001[b] |
| Covariates | | | |
| Randomisation: enhanced prophylaxis | | 0.58 (0.39–0.87) | 0.008 |
| WHO stage at enrolment: 3–4 | | 2.13 (1.27–3.57) | 0.004 |
| Site: Centre B | | 1.89 (0.84–4.24) | 0.33 |
| Centre C | | 0.56 (0.22–1.46) | |
| Centre D | | 1.28 (0.57–2.90) | |
| Centre E | | 1.62 (0.70–3.76) | |
| Centre F | | 1.23 (0.55–2.71) | |
| Centre G | | 1.74 (0.83–3.68) | |
| Age (per 5 years) | | 1.13 (1.03–1.24) | 0.009 |
| CD4 (per 10 cells/mm³) | | 0.89 (0.83–0.95) | 0.001 |
| BMI (per kg/m²) | | 0.92 (0.85–1.01) | 0.07 |
| HIV viral load (per log 10) | | 1.12 (0.78–1.61) | 0.53 |

Note: there was no evidence of interaction between enhanced prophylaxis randomisation and any biomarker or covariate (p > 0.1).

[a]Adjusted models included one biomarker adjusted for all factors under 'Covariates' above. The fully adjusted model included all biomarkers shown together as well as the covariates.

[b]35 biomarkers considered (excluding stool biomarkers and those where >40% of values were outside limit of detection) using backwards elimination (exit p = 0.1, see Methods): naïve Bonferroni significance threshold = 0.05/35 = 0.0014; symbol indicates tests passing an ordered Benjamini–Hochberg (BH) threshold.

the variation (Supplementary Fig. 3). The cluster analysis using these components found four clusters, with 38 biomarkers showing very strong evidence of variation across the clusters (p < 0.001, 16 shown in Fig. 2). Group 1 (n = 264) had relatively low levels of RANTES, IP-10, stromal cell-derived factor 1 (SDF1a), and growth-regulated alpha protein (GROA). Group 2 (n = 77) was characterised by high levels of IL-2, IL-6, granulocyte-macrophage colony-stimulating factor (GM-CSF), and IFN-γ. Group 3 (n = 41) was characterised by high RANTES, IP-10, SDF1a, and eotaxin; all participants in this group were from centres in

one country. Group 4 (n = 203) generally had low concentrations of inflammatory markers, including CRP, IL-6, TNFα, and IL-8. The baseline characteristics of these four groups are shown in Fig. 3. Groups 2 and 3 had significantly lower BMI than the other groups, while other markers of baseline disease severity were similar between groups. Mortality was lower in group 4 (19%) compared to groups 1, 2 and 3 (32%, 32% and 37%, respectively; P = 0.007). Causes of death were broadly similar across all 4 groups. Taken together, our clustering approach identified four broad patterns of biomarkers, but only one associated with lower mortality, which was characterised by having the lowest concentrations of inflammatory biomarkers.

### Enhanced antimicrobial prophylaxis modulates enteropathy in advanced HIV

We finally evaluated the effect of enhanced prophylaxis on early changes in biomarkers from ART initiation to 4 weeks later (Table 3), hypothesising that reduction in infections, together with the immunomodulatory properties of azithromycin, would reduce inflammation and enteropathy, noting that power is lower to detect heterogeneity between enhanced and standard prophylaxis in changes from baseline. The strongest evidence was for enhanced prophylaxis being associated with larger effects than cotrimoxazole alone on the change in plasma I-FABP (interaction $p = 0.002$), faecal MPO ($p = 0.005$) and faecal alpha-1 antitrypsin (A1AT; $p = 0.01$) (none meeting Benjamini–Hochberg threshold accounting for multiple testing). I-FABP increased in both groups between week 0 and 4, but the increase was greater with enhanced prophylaxis (+0.8 $\log_2$ increase; 95% CI (0.7, 0.9) versus +0.5 (0.3, 0.7) with cotrimoxazole alone; difference 0.3 $\log_2$ (0.1, 0.5); interaction $p = 0.002$). Those receiving enhanced prophylaxis showed a greater reduction in faecal MPO (−1.5 $\log_2$ (−2.0, −1.0)) versus cotrimoxazole alone (−0.7 $\log_2$ (−1.1, −0.2); difference −0.8 (−1.4, −0.3); interaction $p = 0.005$). A1AT showed little change to week 4 (+0.0 $\log_2$ (−0.2, 0.3)) with cotrimoxazole alone, but reduced with enhanced prophylaxis (−0.4 (−0.6, −0.1); difference −0.4 $\log_2$ (−0.7, −0.1); interaction $p = 0.01$). Taken together, these data suggest that enhanced prophylaxis had no effect on systemic inflammatory or immunoregulatory biomarkers but did modulate HIV enteropathy during the first few weeks after ART initiation.

### Discussion

Changes in the expression of immune cell-derived soluble factors are associated with disease progression and outcomes in HIV infection[14]. In the current study, we examined a range of inflammatory, immunoregulatory and enteropathy markers, to explore associations with mortality in participants initiating ART with advanced HIV in three African countries. Our study has four major findings. First,

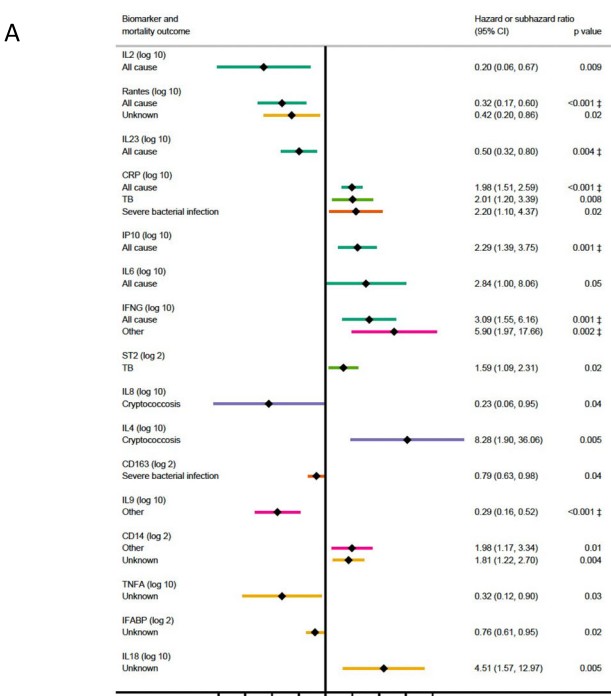

A

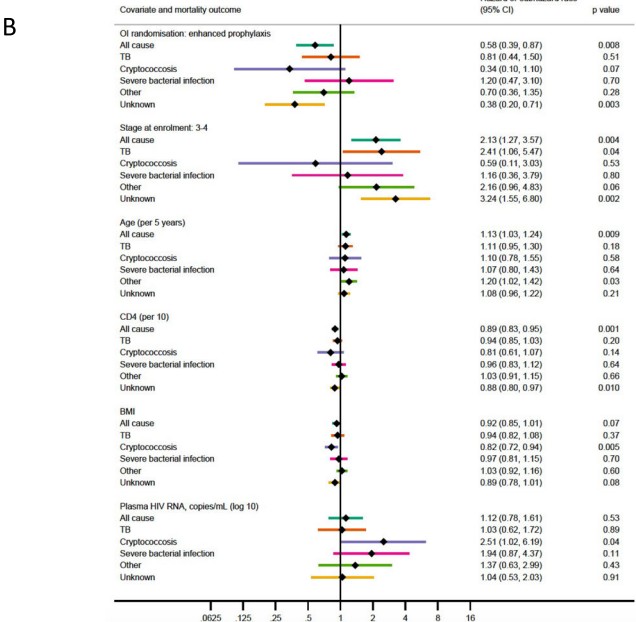

B

**Fig. 1 | Effects of baseline biomarkers on 24 week all-cause and cause-specific mortality in adults and adolescents with CD4 < 100 cells/mm³ initiating antiretroviral therapy in sub-Saharan Africa.** Plots show associations between baseline biomarker concentrations (**A**, top panel) and model covariates (**B**, bottom panel) and all-cause mortality or cause-specific mortality. The model for all-cause mortality shows hazard ratios from a Cox model, while the model for cause-specific mortality shows sub-hazard ratios from Fine and Gray models, with associated two-sided *p*-values. The error bars show 95% confidence intervals, the centre is the point estimate of the hazard or sub-hazard ratio. ‡ 35 biomarkers considered for each cause-specific model (excluding stool biomarkers and those where >40% of values were outside limit of detection) using backwards elimination (exit *p* = 0.1, see Methods): naïve Bonferroni significance threshold = 0.05/35 = 0.0014; symbol indicates tests passing an ordered Benjamini–Hochberg (BH) threshold. All cause model *N* = 582, TB *N* = 591, cryptococcosis *N* = 582, severe bacterial infection *N* = 591, other *N* = 582, unknown *N* = 582. Exact *p*-values: RANTES all-cause 0.0004, CRP all-cause 0.0000007, IL9 other 0.00003. Source data are provided as a Source Data file.

adjunctive antimicrobials to modulate the gut environment in advanced HIV.

In this study, increases in classical inflammatory markers such as C-reactive protein (CRP) and interleukin 6 (IL-6) were associated with all-cause mortality, consistent with the existing literature[15,16]. We also found that other inflammatory biomarkers, which have been studied less frequently in previous cohorts, were independently associated with mortality. Higher levels of interferon gamma-inducible protein 10 (IP-10), an inflammatory chemokine previously associated with severity of respiratory infections[17] and HIV disease progression[18,19], was associated with all-cause mortality in this cohort. Interferon gamma (IFN-γ), a pro-inflammatory cytokine essential for antiviral defence, was also strongly associated with all-cause mortality. IFN-γ has previously been shown to be elevated in HIV infection[20] and has been associated with mortality in other viral diseases such as COVID-19[21], but has not, to our knowledge, previously been shown to be associated with mortality in HIV infection.

Homoeostatic and adaptive immune markers were associated with reduced mortality in this study. In particular, higher levels of IL-2, a pleiotropic cytokine which plays a key role in T-cell homoeostasis and survival, was associated with reduced all-cause mortality. IL-2 is a strong inducer of CD4+ T-cells; the SILCAAT and ESPRIT trials previously showed that adjunctive recombinant IL-2 treatment can effectively increase CD4 counts in people living with HIV, but these gains did not translate into reductions in opportunistic infections or deaths in those trials, primarily because higher CD4 counts were due to existing cells living longer rather than being generated de novo[22]. RANTES (also known as CCL5), shown in this study to be associated with reduced all-cause mortality and deaths from unknown causes, is critical for effective antiviral CD8 T-cell function during chronic viral infections[23]. Finally, IL-23, a proinflammatory cytokine in the IL-12 family, was associated with reduced all-cause mortality. IL-23 confers protection against fungal and bacterial infections, although the specific role of IL-23 in HIV pathogenesis and progression needs to be further examined.

Additionally, we found no statistically significant association between several biomarkers and all-cause mortality, although they were associated with disease-specific causes of death or with unknown/other causes. Interestingly, increases in the Th2 cytokine IL-4 were strongly associated with deaths from cryptococcal disease. Cryptococcal clearance has previously been associated with increased concentrations of Th1 cytokines (IL-12, TNFα) and decreased Th2 cytokines (IL-4, IL-5, IL-12)[24]. Conversely, IL-8/CXCL8, a chemokine involved in neutrophil chemotaxis, was associated with reduced cryptococcal deaths, consistent with the role of neutrophils in anti-fungal defences. Higher soluble serum suppression of tumorigenicity 2 (sST2), a receptor of IL-33 in the IL-1 family, was positively associated

pro-inflammatory and immunoregulatory markers prior to ART initiation were associated with mortality, independently of clinical and immunological disease stage. Second, we found disease-specific baseline patterns of biomarkers associated with mortality from different underlying causes, with distinct signatures suggesting different mechanisms underpinning mortality. Third, by clustering participants based on baseline biomarkers, we could identify four groups with distinct distributions of biomarkers and differential clinical outcomes. Finally, an enhanced prophylaxis package (containing antibacterial, anthelminthic, antifungal and anti-mycobacterial agents) reduced markers of enteropathy in the first few weeks of ART but not systemic inflammation. Collectively, our findings highlight the importance of the immune and inflammatory milieu in determining outcomes following ART initiation, and the potential for

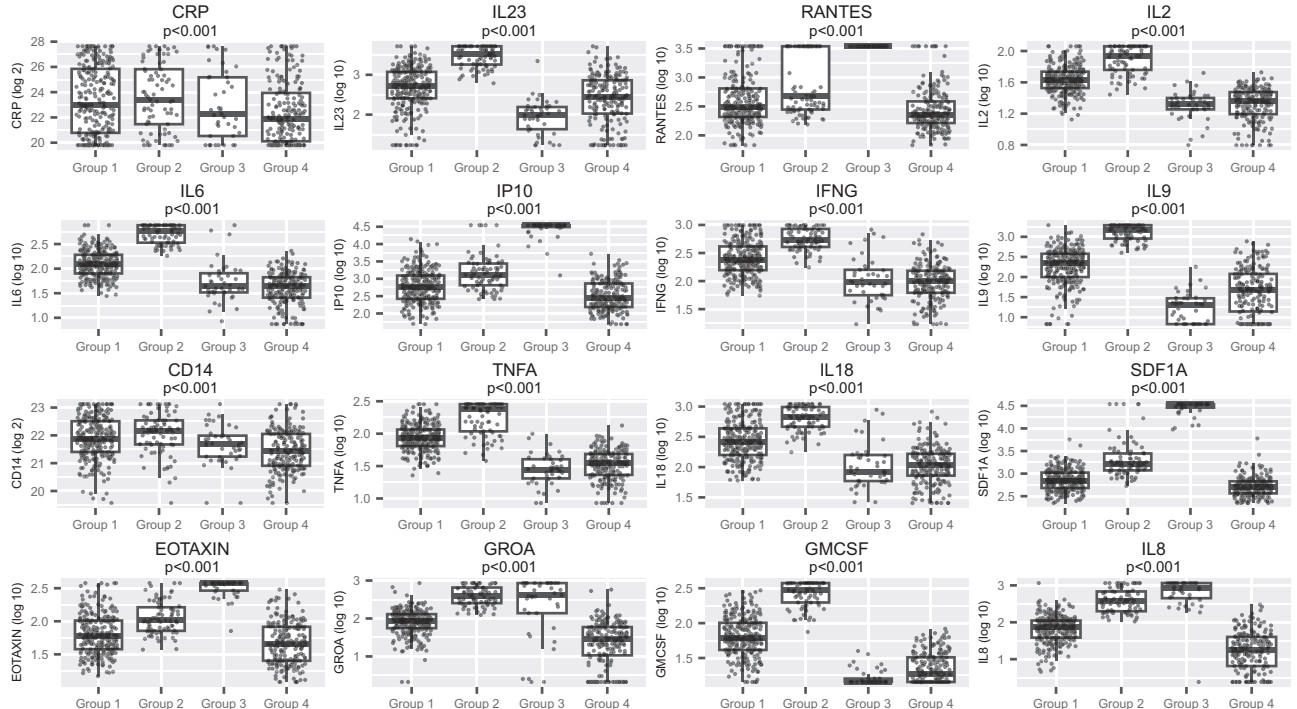

**Fig. 2 | Distribution of key baseline biomarkers in 4 sub-groups identified through hierarchical clustering of principal components of 23 baseline biomarkers.** Hierarchical clustering was undertaken following a principal components analysis, which included all biomarkers, CD4 and viral load, with variables standardised before analysis. The top principal components were used in the hierarchical cluster analysis using Ward's linkage, with the number of clusters determined by the Calinski-Harabasz stopping rule. Box plots show the biomarker distributions within the four clusters identified. The boxes show the 25th and 75th percentiles, and the central line marks the median value. The whiskers extend to the most extreme value no further than 1.5*IQR from the 25th/75th percentile. Each individual value is plotted as a dot. Values above the limit of detection were set at that limit; for example, all RANTES values in group 3 were at the limit of detection. *N* = 585. *P*-values from Kruskal–Wallis tests. All values presented are below Benjamini–Hochberg (BH) threshold. Source data are provided as a Source Data file.

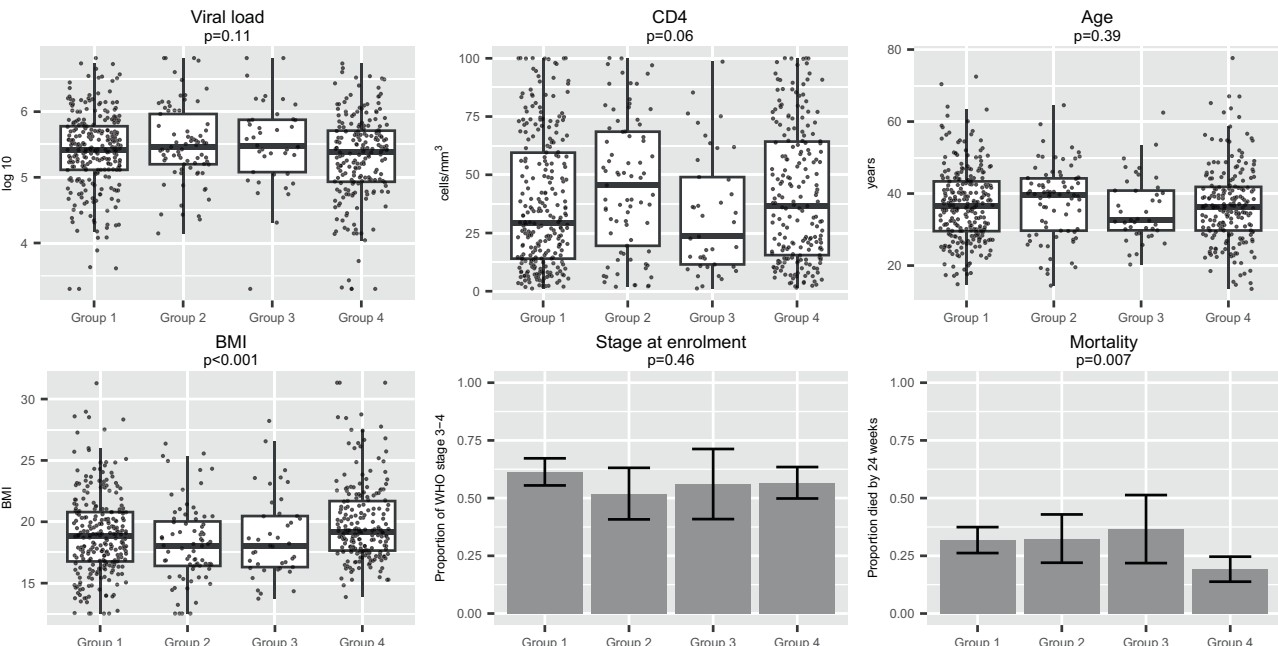

**Fig. 3 | Baseline viral load, CD4, age, BMI, stage and mortality in 4 sub-groups identified through hierarchical clustering.** Among participants in the 4 sub-groups identified through hierarchical clustering of principal components derived from 23 baseline biomarkers, the baseline viral load, CD4 count, age, BMI, WHO disease stage and mortality are shown. Box plots for viral load, CD4 count, age and BMI show the distributions within the four clusters identified. The boxes show the 25th and 75th percentiles, and the central line marks the median value. The whiskers extend to the most extreme value no further than 1.5*IQR from the 25th/75th percentile. Individual data points are plotted as a dot. Bar charts for stage at enrolment and mortality (binary variables) show the means with error bars representing 95% confidence intervals. *N* = 579 for BMI, *N* = 585 for other variables. *P*-values from Kruskal–Wallis tests except stage and mortality which are chi-squared tests with no adjustment for multiple testing. BMI exact *p*-value = 0.0004. Source data are provided as a Source Data file.

**Table 3 | Biomarkers over the first 4 weeks on ART by randomised enhanced prophylaxis vs cotrimoxazole only**

| Variable | Enhanced prophylaxis – mean (+/−SD)[b] | | Cotrimoxazole prophylaxis only – mean (+/−SD)[b] | | Interaction *p*-value[a] |
|---|---|---|---|---|---|
| | Week 0 | Week 4 | Week 0 | Week 4 | |
| *Plasma biomarkers* | | | | | |
| sCD14 (ng/ml) | 3523 (2010–6175) | 3756 (2133–6612) | 3558 (2081–6084) | 3614 (2059–6344) | 0.67 |
| sCD163 (ng/ml) | 935 (403–1688) | 956 (406–1738) | 871 (346–1636) | 908 (346–1737) | 0.57 |
| CRP (mg/l) | 2 (1–4) | 3 (2–4) | 2 (1–4) | 3 (2–4) | 0.84 |
| D-dimer (ng/ml) | 3414 (1422–8197) | 3286 (1402–7701) | 3383 (1259–9089) | 3317 (1280–8601) | 0.65 |
| Eotaxin | 69 (29–163) | 61 (26–144) | 68 (29–155) | 59 (26–137) | 0.14 |
| GM-CSF | 48 (18–128) | 45 (17–118) | 49 (18–133) | 48 (18–126) | 0.78 |
| GROA | 67 (16–282) | 64 (15–276) | 66 (15–293) | 62 (14–272) | 0.43 |
| **I-FABP** | **2098 (892–4933)** | **3433 (1682–7006)** | **1807 (792–4124)** | **2964 (1408–6236)** | **0.002** |
| IFNα | 1.6 (0.4–6.3) | 0.8 (0.3–2.1) | 1.6 (0.4–6.4) | 0.7 (0.3–1.9) | 0.49 |
| IFNγ | 193 (75–494) | 149 (54–413) | 183 (69–481) | 161 (58–444) | 0.67 |
| IL-10 | 12 (3–41) | 11 (3–42) | 11 (3–40) | 11 (3–41) | 0.98 |
| IL-12p70 | 20 (10–41) | 19 (10–38) | 20 (10–41) | 20 (10–39) | 0.41 |
| IL-13 | 11 (5–26) | 11 (5–25) | 11 (5–27) | 11 (5–26) | 0.92 |
| IL-15 | 7 (2–28) | 7 (2–26) | 7 (2–28) | 7 (2–27) | 0.45 |
| IL-17A | 16 (7–37) | 15 (6–36) | 15 (6–36) | 15 (6–34) | 0.46 |
| IL-18 | 210 (82–538) | 162 (60–441) | 197 (76–507) | 172 (63–470) | 0.73 |
| IL-1α | 1.27 (0.41–3.94) | 1.23 (0.40–3.82) | 1.32 (0.41–4.30) | 1.24 (0.40–3.79) | 0.76 |
| IL-1β | 46 (14–147) | 42 (13–136) | 42 (13–139) | 42 (14–132) | 0.83 |
| IL-1RA | 220 (31–1532) | 140 (23–864) | 218 (30–1587) | 159 (24–1067) | 0.90 |
| IL-2 | 35 (19–65) | 33 (18–60) | 33 (18–62) | 33 (18–61) | 0.75 |
| IL-21 | 135 (36–504) | 129 (35–474) | 139 (37–527) | 141 (38–523) | 0.98 |
| IL-22 | 167 (47–596) | 155 (44–548) | 164 (45–603) | 162 (46–573) | 0.69 |
| IL-23 | 421 (96–1839) | 422 (96–1855) | 421 (89–2003) | 456 (106–1963) | 0.79 |
| IL-27 | 106 (28–399) | 99 (27–368) | 100 (26–387) | 102 (27–389) | 0.74 |
| IL-31 | 44 (7–274) | 36 (6–211) | 43 (7–265) | 38 (7–219) | 0.89 |
| IL-4 | 45 (10–203) | 42 (10–181) | 48 (11–209) | 47 (11–196) | 0.84 |
| IL-5 | 60 (24–149) | 52 (21–129) | 57 (24–137) | 54 (23–129) | 0.59 |
| IL-6 | 96 (33–273) | 85 (29–248) | 95 (32–281) | 89 (30–267) | 0.48 |
| IL-7 | 3.7 (1.1–12.6) | 3.0 (0.9–10.3) | 3.5 (1.0–12.2) | 2.8 (0.8–9.3) | 0.40 |
| IL-8 | 60 (13–273) | 47 (10–235) | 60 (12–285) | 50 (10–253) | 0.48 |
| IL-9 | 118 (24–586) | 107 (21–538) | 123 (24–635) | 124 (25–620) | 0.66 |
| IP10 | 604 (177–2065) | 423 (116–1544) | 623 (193–2014) | 438 (123–1556) | 0.80 |
| LBP (mg/l) | 26 (17–39) | 23 (14–38) | 27 (17–44) | 24 (14–41) | 0.95 |
| MCP1 | 96 (43–369) | 77 (35–286) | 100 (46–364) | 76 (36–248) | 0.06 |
| MIP1α | 13 (3–46) | 12 (3–45) | 12 (3–45) | 11 (3–40) | 0.06 |
| MIP1β | 140 (40–491) | 140 (41–479) | 134 (37–480) | 129 (38–438) | 0.20 |
| RANTES | 450 (136–1482) | 461 (138–1543) | 469 (149–1475) | 440 (142–1367) | 0.24 |
| SDF1A | 939 (292–3021) | 883 (278–2801) | 957 (290–3157) | 857 (269–2728) | 0.29 |
| sST2 | 14181 (5400–37238) | 11927 (4604–30898) | 15250 (5423–42888) | 13441 (4612–39170) | 0.68 |
| TNFα | 65 (29–145) | 58 (27–128) | 60 (28–130) | 58 (27–127) | 0.75 |
| TNFβ | 29 (5–164) | 28 (5–156) | 34 (5–225) | 32 (5–193) | 0.53 |
| *Stool biomarkers* | | | | | |
| **A1AT (mg/l)** | **259 (100–673)** | **200 (80–496)** | **268 (100–718)** | **262 (114–599)** | **0.01** |
| **MPO (ng/ml)** | **2168 (804–5843)** | **1258 (590–2683)** | **2056 (760–5560)** | **1664 (652–4248)** | **0.005** |
| NEO (nmol/l) | 270 (60–1216) | 569 (136–2376) | 350 (76–1614) | 664 (159–2770) | 0.86 |

Values given in pg/ml unless otherwise stated.

[a]*P*-values for interaction of between enhanced prophylaxis randomisation and time from mixed model adjusted for baseline CD4 assess whether there is evidence that the changes in biomarker from week 0 to 4 differ between the randomised prophylaxis groups. 44 biomarkers tested: naïve Bonferroni significance threshold = 0.05/44 = 0.00114; no test meets the ordered BH threshold. Variables with *P*-value < 0.05 in bold.

[b]Mean and standard deviation calculated on scale transformed for normality for the mixed model and then back transformed onto the original scale for interpretability.

with TB and with deaths from other causes. The IL-33/ST2 axis is emerging as a key player in induction of innate and adaptive immune responses, especially at epithelial barriers, and in tissue remodelling[25]. sST2 has been previously shown to be associated with all-cause mortality in adults living with HIV[26], and is positively correlated with CD8 counts, activation and exhaustion in early HIV infection[27]. To our knowledge this is the first study to show specific associations with TB deaths, but there is an emerging interest in the role of the IL-33/ST2 axis in protection against TB[28]. Despite review by an endpoint committee, a substantial fraction of deaths did not have a cause identified (unknown causes) in this cohort of complex, sick patients with advanced HIV, many dying suddenly at home. IL-18, a proinflammatory cytokine in the IL-1 family, associated with deaths from unknown causes in this study, has previously been associated with cardiovascular deaths[29–31]. TNFα, a proinflammatory cytokine with broad roles in host defence against viral, bacterial and parasitic infections[32], was also associated with a protective effect against deaths from unknown causes in this study. High sCD14, a soluble receptor released primarily by activated monocytes and macrophages which binds LPS, was associated with deaths from a range of other causes. sCD14 has previously been shown to be independently associated with all-cause mortality in advanced HIV and was strongly correlated with other inflammatory markers in this study, such as IL-6, CRP and D-dimer[33]. By contrast, monocyte activation, indicated by elevated sCD163, a soluble form of the monocyte- and macrophage-specific scavenger receptor, was associated with protection from deaths due to severe bacterial infections in this study. Monocytes have a central role in microbial detection through pattern recognition receptors, and in antibacterial defence via reactive oxygen intermediates and phagolysosome enzymes[34,35]. Previously, sCD163 has been shown to be associated with all-cause mortality in ART-naïve HIV-infected individuals[36], in contrast to the current study. These discrepancies may highlight the fine balance that exists in the setting of HIV infection between protective anti-inflammatory responses to co-infections, and disease progression due to exuberant inflammation.

Advanced HIV leads to alteration in gut structure and function, due to loss of CD4 (particularly Th17) cells in the lamina propria[37], alterations in the microbiome[38], and damage to the protective intestinal barrier which usually maintains gut integrity[39]. We did not find any independent associations between the baseline severity of enteropathy (as measured by I-FABP, faecal myeloperoxidase, and alpha-1 antitrypsin) and mortality, similar to some[40] (but not all[33]) previous studies. However, the randomised antimicrobial bundle (enhanced prophylaxis) led to greater changes in these biomarkers over the first 4 weeks on ART, compared to standard-of-care cotrimoxazole alone. It is plausible that azithromycin and albendazole reduced the burden of enteropathogens (including parasitic worms), which may drive enteropathy. Furthermore, azithromycin has well-recognised immunomodulatory effects[41], and has previously been shown to reduce intestinal inflammation in Indian infants[42]. A combination of antimicrobial and anti-inflammatory activity could have led to the modulation of enteropathy soon after ART initiation, whereas enhanced prophylaxis did not have any effects on systemic inflammatory markers.

Our study had strengths and limitations. We leveraged a large cohort of participants with advanced HIV including both adolescents and adults together in the same study, thereby increasing its generalisability. We used rich data on causes of death, which were independently adjudicated by blinded reviewers, allowing us to specifically investigate cause-specific biomarker signatures; however, the pragmatic nature of the trial meant that causes of death could not be identified in a substantial minority. We were able to use baseline samples from all participants who died, providing us with sufficient power to identify important associations with biomarkers at ART initiation. We measured a wide range of plasma and faecal markers,

which led to identification of novel associations. We used data reduction techniques to handle multiple analytes and to identify clustering of biomarkers, and we validated our selected markers using best subsets regression. However, our choice of multiple markers may have increased the risk of type 1 error. We did not adjust $p$-values directly for multiple comparisons, as this study consisted of post-hoc exploratory analyses. Rather we interpreted findings in the context of the number of tests performed, and focussed on assessing consistency across the different analyses. Standard methods for adjustment are conservative when comparisons are not independent, as is the case here since many of the biomarkers are correlated.

In summary, several soluble inflammatory, homoeostatic and adaptive immune markers at ART initiation are associated with mortality following ART initiation, with distinct biomarker patterns depending on cause of death. Further studies exploring the pathways indicated by markers associated with specific causes of death may help to identify the root drivers or mechanisms and offer insights into possible cause-specific interventions. However, enhanced antimicrobial prophylaxis modulates enteropathy in this cohort of participants with advanced HIV infection, as well as independently reducing all-cause mortality. Whether the changes in enteropathy biomarkers directly contribute to clinical benefits, such as through improved nutrient and drug absorption, also warrants further study.

## Methods
### REALITY trial
The Reduction of Early mortality (REALITY) trial (ISRCTN43622374) was undertaken between 2013 and 2016 in Kenya, Malawi, Uganda and Zimbabwe and recruited ART-naïve HIV-infected adults and children who were 5 years or older with a CD4 count <100 cells per mm³. Participants of any sex or gender were eligible to enrol. Participants were randomised at ART initiation to three interventions in a $2 \times 2 \times 2$ factorial design: enhanced antimicrobial prophylaxis, adjunctive raltegravir therapy, and ready-to-use supplementary food[12,43,44]. The bundle of enhanced infection prophylaxis comprised continuous cotrimoxazole plus at least 12 weeks of isoniazid/pyridoxine as a single fixed-dose combination tablet, 12 weeks of fluconazole, 5 days of azithromycin, and single-dose albendazole. Patients in the standard-of-care arm received cotrimoxazole alone. The intervention bundle conferred a 27% relative reduction in mortality by 24 weeks (primary trial outcome), and 24% mortality reduction by 48 weeks, as previously reported[12]. All deaths were reviewed by an endpoint review committee with independent chair, to adjudicate cause of death, during the trial (i.e. without knowledge of levels of biomarkers generated subsequently on stored samples).

### REALITY sample collection and storage
Blood was drawn into EDTA tubes and processed within 2 h at the screening and baseline visits, then at weeks 4, 12, 24, 36 and 48 post-randomisation. Plasma was separated from cells by centrifugation. The buffy coat cell layer was collected and treated with FACS Lysing Solution (BD Biosciences, San Jose, CA) to lyse red blood cells and fix leucocytes prior to freezing in a mixture of DMSO, fetal calf serum and phosphate-buffered saline, as previously described[45]. Stool was collected by participants into a plain container prior to scheduled clinic visits (in Harare and Kilifi only) at baseline, 4, 12 and 48 weeks, and transferred using a spatula into a plain storage vial. All samples were stored at −80 °C.

### Immunology substudy
This study used a case-cohort design in the study centres in Kenya, Uganda and Zimbabwe (ethical approval could not be obtained for the substudy in Malawi as the trial had closed). Random sampling stratified by site (see below) was first done across a combined group comprising all deaths occurring by 24 weeks (the vast majority of deaths through

48 weeks (trial duration), 169/201 (84%)) and those who remained in follow-up (i.e., alive) until 48 weeks with complete sets of samples to week 24 and data on baseline CD8⁺ T-cell counts (the vast majority of those not known to have died, 1060/1316 (81%)) (Supplementary Fig. 1). The case-cohort design first randomly sampled 45% of participants from sites storing stool, buffy coat cells, plasma and baseline cell pellet (90% from one of these two sites because of missing baseline CD8⁺ due to reagent unavailability), 45% from sites storing buffy coat cells and plasma/cell pellets (but not stool), and 10% from one single site storing plasma/cell pellets only. Sampling was also stratified by CD4 count (0–24, 25–49, 50–99 cells/mm³; approximate terciles). Any deaths by 24 weeks not selected by the random sampling were added to the sample so that the final sample (total $N = 599$) included all 169 deaths occurring by 24 weeks. The study focused on early changes in each pathway (from baseline to 4 weeks) since the enhanced prophylaxis bundle was only given for the first 12 weeks and most deaths occurred early (prior to 24 weeks)[7]. All available baseline and 4 weeks post-ART initiation samples were therefore retrieved and assayed by laboratory scientists who were blinded to trial arm and clinical outcomes. Since this is an exploratory analysis, which is a sub-study of a randomised trial, we did not pre-specify an analysis stratified by sex or gender.

### Enteropathy biomarkers
Intestinal fatty acid binding protein (I-FABP) is a marker of small intestinal enterocyte damage. Plasma concentrations were measured by ELISA according to the kit manufacturer's instructions (Human FABP2/I-FABP Quantikine ELISA; R&D Systems Inc, Minneapolis, MN, USA). A Quantikine Immunoassay FABP2/I-FABP control set (R&D Systems Inc) was used for assay quality control. Stool samples were tested by ELISA for neopterin (limit of detection (LOD) 0.7 nmol/L; GenWay Biotech Inc, San Diego, CA, USA), myeloperoxidase (LOD 1.6 ng/mL; Immundianostik, Bensheim, Germany), and alpha-1 anti-trypsin (A1AT; LOD 1.5 ng/mL; BioVendor, Brno, Czech Republic). Biomarker concentrations were determined against a standard curve; samples above the upper LOD were re-run at lower dilutions.

### Inflammatory and immunoregulatory biomarkers
A preconfigured ProcartaPlex 34-plex Human cytokine & chemokine panel (ThermoFisher Scientific/Life Technologies Ltd) was used to assess the concentrations of secreted proteins in plasma (Eotaxin/CCL11; GM-CSF; GROA/CXCL1; IFNα; IFN-γ; IL-1β; IL-1α; IL-1RA; IL-2; IL-4; IL-5; IL-6; IL-7; IL-8/CXCL8; IL-9; IL-10; IL-12p70; IL-13; IL-15; IL-17A; IL-18; IL-21; IL-22; IL-23; IL-27; IL-31; IP-10/CXCL10; MCP-1/CCL2; MIP-1α/CCL3; MIP-1β /CCL4; RANTES/CCL5; SDF1α/CXCL12; TNFα; TNFβ/LTA). Two customised 3-plex Luminex assays (R&D Systems Inc, Minneapolis, MN, USA) were used for detection of plasma C-reactive protein (CRP), lipopolysaccharide binding protein (LBP), soluble CD14 (Panel 1), D-dimer, soluble CD163 and soluble ST2 (Panel 2) based on the recommended dilution of the analytes in plasma. All multiplex assays were run in singlicate on a Luminex MagPix machine with xPonent 4.2 software. Biomarker concentrations were determined against the respective standard curves and samples above the upper limit of detection were re-run at lower dilutions.

### Sample size
The target sample size for plasma baseline biomarkers was 602 patients (actual 599 due to natural variability in the random sampling). This provided 80% power to detect a hazard ratio for mortality associated with each biomarker quartile of 0.78, adjusting for the case-cohort design[46].

### Statistical analysis
Statistical analyses were conducted in Stata 16.1, and Figs. 2 and 3 created in R version 4.2.2. Stata analysis code is available in Supplementary Data 1, and data used to produce all figures are available in Supplementary Data 2. Biomarker values were truncated at the 1st and 95th percentile, since most data were right-skewed with high outliers. Values above or below the limits of detection (LOD) were set to the LOD. Associations between baseline values and all-cause mortality at 24 weeks were analysed using a Cox model. Thirty-five biomarkers were included as candidate covariates, and backwards elimination with exit $p = 0.1$ was used for variable selection. Biomarkers selected in the model were tested for interactions with the randomisation to enhanced prophylaxis. Similarly, cause-specific mortality was analysed for five causes (TB, cryptococcosis, severe bacterial infection (SBI), other and unknown; deaths could be from multiple causes) using Fine & Gray models[47], with death for another cause treated as a competing risk.

All models were adjusted for prophylaxis randomisation, viral load, CD4, WHO stage, age and BMI at enrolment, and centre; and weighted according to inverse probability of selection into the sub-study. Deaths were weighted as 1, and non-deaths were weighted (by centre) to reflect the inverse probability of selection from all REALITY patients at each centre ≥13 years of age and alive at week 48 (regardless of immunology substudy membership and available samples) in order to represent this population. All biomarkers were modelled as continuous variables on the $\log_{10}$ scale for comparability across biomarkers; there was no evidence of non-linearity (assessed using fractional polynomials). For each specific cause of death, and for all-cause mortality, best subsets regression was used to compare the model selected through backwards elimination to other candidate models. All possible models with the same number of covariates as the model selected through backwards elimination were fitted to the data and compared using AIC.

To identify whether there were underlying subgroups (or "phenotypes") of participants based on baseline biomarker combinations, principal components analysis was run on all biomarkers, CD4 and viral load, with variables standardised before analysis. The number of principal components was chosen by examining the scree plot with an aim of explaining around 80% of the variation (Supplementary Fig. 3). The top principal components were used in a hierarchical cluster analysis using Ward's linkage, with the number of clusters determined by the Calinski-Harabasz stopping rule. Box plots were used to describe biomarker distributions within the clusters.

Mean values of biomarkers (transformed $\log_{10}$, except I-FABP, sST2, D-dimer, sCD14, sCD163, LBP, A1AT, myeloperoxidase and neopterin, which used $\log_2$ transformation) at baseline and week 4 were estimated using mixed models for interval data (the meintreg command in Stata) to account for truncation of values at the LOD. Time, centre, and baseline CD4 were included as fixed effects, and participant as a random effect with unstructured covariance. Models were weighted according to inverse probability of selection into the substudy. The effect of the prophylaxis randomisation on the change in biomarker levels was investigated by including the interaction between the randomisation and time at week 4.

### Ethics and inclusion statement
Researchers from Zimbabwe, Uganda, Kenya and Malawi were involved in the design, implementation, and interpretation of the REALITY trial and were authors on all manuscripts arising from the trial. The trial steering committee included researchers and independent members from each country. Adult participants provided written informed consent, and parents/guardians provided written informed consent for participants below the age of 18 years to enrol in the trial, which included storage of biological specimens for subsequent analysis. Older children additionally provided assent, according to national guidelines. The trial and the laboratory work in this study were approved by ethics committees in Kenya (Moi University Institutional Research and Ethics Committee and the Kenya Medical Research Institute Ethics Review Committee), Uganda (Joint Clinical Research

Centre Institutional Review Board and the Uganda National Council for Science and Technology), Zimbabwe (Joint Parirenyatwa Hospital and College of Health Sciences Research Ethics Committee and the Medical Research Council of Zimbabwe), and the UK (University College London Ethics Committee).

## Reporting summary

Further information on research design is available in the Nature Portfolio Reporting Summary linked to this article.

## Data availability

The full dataset used in the analyses presented in this manuscript are available on Figshare at https://doi.org/10.6084/m9.figshare.24459760. Source data are provided with this paper.

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

## Acknowledgements

Funded by MRC (grant MR/P022251/1) and Wellcome (108065/Z/15/Z to A.J.P.). We thank the participants, collaborating sites, and the REALITY trial team.

## Author contributions

V.R. contributed laboratory analysis, data interpretation and manuscript drafting. R.C. contributed data analysis, data interpretation and manuscript drafting. A.G. contributed specimen storage, laboratory analysis and manuscript revision. J.M. contributed specimen storage, laboratory analysis and manuscript revision. R.N. contributed specimen storage, laboratory analysis and manuscript revision AS contributed study oversight, participant enrolment, specimen collection and manuscript revision. M.B.-D. contributed study oversight, participant enrolment, specimen collection and manuscript revision. V.M. contributed study oversight, participant enrolment, specimen collection and manuscript revision. J.A.B. contributed study oversight, data interpretation and manuscript revision. A.J.S. contributed data analysis, data interpretation and manuscript drafting. D.M.G. contributed study leadership, data interpretation and manuscript revision. A.S.W. contributed study leadership, data analysis, data interpretation and manuscript writing. N.K. contributed data interpretation and manuscript writing. A.J.P. contributed study leadership, laboratory oversight, data interpretation and manuscript writing.

## Competing interests

The authors declare no competing interests.
