## [Peer Review File · Nature Communications]

REVIEWER COMMENTS

Reviewer #1 (Remarks to the Author):

Overall, the statistical approach is clearly defined and aligns with the hypotheses. Conclusions are well supported by the analyses.

1. I appreciate the clear statement of well-defined hypotheses early in the manuscript. It would be helpful to state which statistical approaches were used for each of the hypotheses in the Methods section. For example, it is unclear what question the principal components analysis was designed to address.

2. Line 310: The text here implies this was not a true case cohort sample but a case control sample. This does not match with the rest of the description.

3. Line 352: Why truncate at 1st and 95th as opposed to a more symmetric truncation (eg 1st and 99th)?

4. Please include a reference for the Fine-Gray model and the mixed models for interval data.

5. Please reference the statistical software (with version) used.

6. Did the Endpoint Adjudication Committee use any of these biomarkers to help make their determination about cause of death?

7. Given that the study was not powered for interactions, is it fair to say "the association between biomarker levels and mortality was not modified by the intervention?" I'm assuming, based on the text, only the p-value was examined. Also, the p-value is from test on means, but the table reports medians and IQRs. It would be important to report the means for all to enhance interpretation. I see these are in the text for significant ($p < 0.05$?) interactions.

8. Was there any adjustment for multiple comparisons?

Reviewer #2 (Remarks to the Author):

The authors investigated associations between baseline immune biomarkers and mortality by 24 weeks since enrollment in 599 participants with advanced HIV who were on ART. The paper is clearly written and addresses important questions. The survival analysis including the competing risk modeling with weights seems solid. Here are a few comments.

- 1) The relationship between the 599 participants with biomarkers measured and the “sub-study” participants who were randomized to the 2 ART treatment groups is unclear. Are they the same cohort? If yes, please clarify. If not, it'll be helpful to include a consort diagram.
- 2) In the description of the immunology substudy, it is unclear whether a case-control design or a case-cohort design was used. It was described as a case-cohort design, but it appeared that the sampling of was depended on the case/non-cases status. Please clarify.
- 3) A total of 23 baseline markers were examined. In the presentation of the analyses results where the effect of each of these markers was examined (adjusted for VL, CD4 etc.), multiple comparison adjustment should be considered for example in Table 1, Figure 1 etc.
- 4) The AIC model selection procedure may not be the best way to illustrate the relative utility of the baseline markers. The authors are encouraged to consider variable importance approaches.
- 5) The clustering analyses using the first 8 PCA components is somewhat arbitrary. They together explained about 80% of variation. How much variation was explained by the first 2, for example? Figure S2 is really difficult to view, but appeared that the majority of the markers had similar/equal load in PC1? The identification of the 4 clusters seems vague as well. Please clarify.
- 6) A minor comment: in Table 1, when different n's are listed for each biomarker, somewhere in the legend or text should explain why.

Reviewer #3 (Remarks to the Author):

This study examined a range of inflammatory, immunoregulatory and enteropathy markers, and sought to define their associations with mortality in participants with advanced HIV initiating ART and the type of antimicrobial treatment received. A large cohort of 599 participants with 169 that

died and 430 that survived, provides meaningful sample sizes with clear distinctions in well-established differentiators of disease outcome between these two groups. The authors show that participants that received enhanced prophylaxis have with lower mortality rates than those that received or standard of care, in line with their previous findings from a much larger cohort size. Of those that died, a significant proportion were found to be attributed to bacterial and fungal infections (but with other causes that were more generic or unknown).

Baseline levels of an extensive suite of variables were analysed identifying inflammatory mediators be associated with higher mortality rates (CRP, IFN-g, IL-6, IP-10) and immunoregulatory mediators (IL-23, IL-2 and RANTES) with lower mortality rates.

When grouped by cause of death, distinct markers were further identified particularly in TB- and cryptococcosis-associated mortality highlighting disease specific signatures.

Further cluster analysis confirmed reduced inflammatory markers CRP, IL-6, TNF-a and IL-8 to be associated with lower mortality. Broad patterns and greater changes in specific marker expression post-ART treatment, particularly in those that received enhanced prophylaxis suggests reduced enteropathy but not systemic inflammation.

Overall, the study is well-written and presented. These findings not only identify and demonstrate how these markers change and/or are different with respect to the cause of death and prophylactic treatment but also provide important insights into the potential mechanisms at play during these disease states.

Major comments:

1. It would be useful to include a figure summarising the patient cohort, how they were stratified especially with respect to the stratification of the sub-study (enhanced vs standard; was this only in the group that died or all patients?) and type of analyses performed.
2. Figures are of low resolution and in some, almost impossible to make out the labels particularly for the Supplementary figures.
3. There are no figure legends (at least in this reviewer's copy) for any of the main figures or supplementary figures.
4. In all figures, please include individual patient data points for transparency of spread and variation.
5. From Table 2, it is claimed that IL-6 was associated with increased all-cause mortality however p-values from these comparisons was 0.05 and therefore not significant.
6. In the cluster analysis (Fig 2), it is not immediately obvious what mediators are significant amongst groups. While the titles on each graph indicate ($p < 0.001$), without a figure legend and statistic bars, it is difficult to ascertain what is different.

7. With regard to analysing markers in those that received the enhanced vs standard treatment, if it has not been done so (see comment 1), it would be revealing to segregate these groups further into those that survived and those that died given the differences in mortality rates between these treatments.

Minor comments:

- In Table 1, the control group varies between variables examined, i.e. for age, sex, CD4 count, BMI variables, this is assumed to be those who survived but for the enhanced prophylaxis variable, this is presumably the standard-of-care treatment. It would greatly help interpretation of results if these control groups are clearly defined. Perhaps by asterisks/symbols within the table and the compared groups indicated at the bottom.
- Include abbreviation for myeloperoxidase (MPO) in main text.
- In text referring to supplementary Figure 2, from the cluster analysis, it is indicated that IL-7 is low in Group 2 however there is no IL-7 in the figure.
- On the graph, the levels for RANTES for Group 3 appear to be beyond the range of the y-axis. Or are these levels above the detection limit?
- "Mortality was lower in group 4 (19%) compared to groups 1, 2 and 3 (32%, 32% and 37%, respectively; $P=0.007$). Causes of death were broadly similar across all 4 groups." No graphical data for this is presented in the manuscript and would be useful to show as there is space in the figure for it.

REVIEWER COMMENTS

Reviewer #1 (Remarks to the Author):

Overall, the statistical approach is clearly defined and aligns with the hypotheses. Conclusions are well supported by the analyses.

1. I appreciate the clear statement of well-defined hypotheses early in the manuscript. It would be helpful to state which statistical approaches were used for each of the hypotheses in the Methods section. For example, it is unclear what question the principal components analysis was designed to address.

Response: We have now clarified this in the revised Methods section.

2. Line 310: The text here implies this was not a true case cohort sample but a case control sample. This does not match with the rest of the description.

Response: The design was a case cohort sample, but we made a small number of restrictions on those considered for the original sampling in order to focus the comparisons on early deaths (within 24 weeks of starting antiretroviral therapy) and to exclude a small number of individuals whose vital status was unknown at 48 weeks. One site was also unable to perform CD8 cell counts for a period of time and we did not include these individuals in the random sampling, since they would have missing data unrelated to their status. These inclusions/exclusions before the case-cohort random selection were inadvertently presented as cases/non-cases – this has now been clarified, and a flow diagram added to supplementary material as suggested by other reviewers to add understanding of the design.

3. Line 352: Why truncate at 1st and 95th as opposed to a more symmetric truncation (eg 1st and 99th)?

Response: Most of the biomarker data were very right skewed with high outliers, therefore the 95th percentile was chosen. We have clarified this in the statistical analysis section, which reads:

“Biomarker values were truncated at the 1st and 95th percentile, since most data were right-skewed with high outliers.”

4. Please include a reference for the Fine-Gray model and the mixed models for interval data.

Response: We have added a reference for the Fine and Gray model. The mixed model for interval data used Stata’s meintreg command. We have added this to the text.

5. Please reference the statistical software (with version) used.

Response: We have added this to the text:

“Statistical analyses were conducted in Stata 16.1, and Figures 2 and 3 created in R version 4.2.2.”

6. Did the Endpoint Adjudication Committee use any of these biomarkers to help make their determination about cause of death?

Response: No, the Endpoint Review Committee adjudicated causes of death during the main REALITY trial, prior to this laboratory substudy. The committee had access to narrative reports from clinicians, and any clinical laboratory tests or radiology that may have been conducted, but not biomarkers. We have clarified this in the text:

“All deaths were reviewed by an endpoint review committee with independent chair, to adjudicate cause of death, during the trial (i.e. without knowledge of levels of biomarkers generated subsequently on stored samples).”

7. Given that the study was not powered for interactions, is it fair to say "the association between biomarker levels and mortality was not modified by the intervention?" I'm assuming, based on the text, only the p-value was examined. Also, the p-value is from test on means, but the table reports medians and IQRs. It would be important to report the means for all to enhance interpretation. I see these are in the text for significant ($p < 0.05$?) interactions.

Response: We have rewritten this sentence to be clear that there was no evidence from our data that the association was modified by the intervention:

“There was no evidence that the association between biomarker levels and mortality was modified by the enhanced prophylaxis intervention (interaction $p > 0.12$).”

Table 3 has been updated to show means rather than medians.

8. Was there any adjustment for multiple comparisons?

Response: We did not adjust for multiple comparisons because this study contained only post-hoc exploratory analyses, with many correlated outcomes. We have added a paragraph to the discussion to acknowledge this:

“We did not adjust p-values directly for multiple comparisons, as this study consisted of post-hoc exploratory analyses. Rather we interpreted findings in the context of the number of tests performed, and focussed on assessing consistency across the different analyses. Standard methods for adjustment are conservative when comparisons are not independent, as is the case here since many of the biomarkers are correlated.”

Reviewer #2 (Remarks to the Author):

The authors investigated associations between baseline immune biomarkers and mortality by 24 weeks since enrollment in 599 participants with advanced HIV who were on ART. The paper is clearly written and addresses important questions. The survival analysis including the competing risk modeling with weights seems solid. Here are a few comments.

1) The relationship between the 599 participants with biomarkers measured and the “sub-study” participants who were randomized to the 2 ART treatment groups is unclear. Are they the same cohort? If yes, please clarify. If not, it'll be helpful to include a consort diagram.

Response: All immunology substudy participants were drawn from the REALITY trial and were therefore randomized to standard or enhanced prophylaxis. This manuscript therefore uses two approaches: first, an analysis outside of the randomization which explores associations between baseline biomarker values and causes of death; and, second, an analysis that leverages the randomization to explore the effects of enhanced prophylaxis on the biomarker values over time. We have now included a CONSORT diagram which shows how participants were selected into the substudy and how many were in each randomized group.

2) In the description of the immunology substudy, it is unclear whether a case-control design or a case-cohort design was used. It was described as a case-cohort design, but it appeared that the sampling of was depended on the case/non-cases status. Please clarify.

Response: We apologise for the lack of clarity. Please see response to reviewer 1 above – this is now clarified in the revised text.

3) A total of 23 baseline markers were examined. In the presentation of the analyses results where the effect of each of these markers was examined (adjusted for VL, CD4 etc.), multiple comparison adjustment should be considered for example in Table 1, Figure 1 etc.

Response: Please see response to reviewer 1 above. We did not adjust for multiple comparisons because this study contained only post-hoc exploratory analyses, with many correlated outcomes. We have added a paragraph to the discussion to acknowledge this.

4) The AIC model selection procedure may not be the best way to illustrate the relative utility of the baseline markers. The authors are encouraged to consider variable importance approaches.

Response: Variable importance is an approach which is typically used where the goal of model fitting is prediction (estimating the change in accuracy of predictions when a variable is or is not included in a model) – as suggested by the reviewer’s focus on utility. However, here we are not trying to predict mortality, but to assess *associations* with mortality. In practice there are numerous methods that could have been used (including Bayesian Information Criterion, BIC, rather than AIC); we did consider penalised approaches but could not identify a package that would fit these in the time-to-event survival framework with the case-cohort design weights. We did test two different approaches, backwards elimination and best subsets regression, and – as per our discussion on multiple testing – we focus on consistency of findings given all analyses are exploratory and post-hoc. We have clearly described our analyses as associative rather than predictive, so our preference would be not to conduct further work on model selection at this stage, since we are not trying to claim that any one biomarker is most useful.

5) The clustering analyses using the first 8 PCA components is somewhat arbitrary. They together explained about 80% of variation. How much variation was explained by the first 2, for example? Figure S2 is really difficult to view, but appeared that the majority of the markers had similar/equal load in PC1? The identification of the 4 clusters seems vague as well. Please clarify.

Response: The number of PCs was chosen based on a combination of considering where the scree plot levels off, and aiming to explain around 80% of variation. We have added this to the methods section and included these plots in a supplementary figure. We acknowledge this choice is somewhat arbitrary as 9 or 10 components could have been chosen instead. While we used the PCs to define the clusters, comparison of the clusters was based on the original biomarker values not the PCs themselves, and hence it is unlikely that clustering based on 9 or 10 rather than 8 PCs would have made much difference to these, given the small absolute % variation PCs 9 and 10 explain.

We apologise for the figure quality which has been improved in the resubmission. For PC1, the majority of components loaded positively at 0.17-0.20 and this PC represents upregulation of a suite of pro- and anti-inflammatory biomarkers. The number of clusters was based on the Calinski-Harabasz stopping rule (as stated in the original Methods), which identified 4 clusters as the most appropriate number.

6) A minor comment: in Table 1, when different n’s are listed for each biomarker, somewhere in the legend or text should explain why.

Response: We have added the following to the footnote of Table 1: “The number of participants (N) with data for each biomarker is shown and differs due to sample availability or technical failures.”

Reviewer #3 (Remarks to the Author):

This study examined a range of inflammatory, immunoregulatory and enteropathy markers, and sought to define their associations with mortality in participants with advanced HIV initiating ART and the type of antimicrobial treatment received. A large cohort of 599 participants with 169 that died and 430 that survived, provides meaningful sample sizes with clear distinctions in well-established differentiators of disease outcome between these two groups. The authors show that participants that received enhanced prophylaxis have with lower mortality rates than those that received or standard of care, in line with their previous findings from a much larger cohort size. Of those that died, a significant proportion were found to be attributed to bacterial and fungal infections (but with other causes that were more generic or unknown).

Baseline levels of an extensive suite of variables were analysed identifying inflammatory mediators be associated with higher mortality rates (CRP, IFN-g, IL-6, IP-10) and immunoregulatory mediators (IL-23, IL-2 and RANTES) with lower mortality rates. When grouped by cause of death, distinct markers were further identified particularly in TB- and cryptococcosis-associated mortality highlighting disease specific signatures. Further cluster analysis confirmed reduced inflammatory markers CRP, IL-6, TNF-a and IL-8 to be associated with lower mortality. Broad patterns and greater changes in specific marker expression post-ART treatment, particularly in those that received enhanced prophylaxis suggests reduced enteropathy but not systemic inflammation.

Overall, the study is well-written and presented. These findings not only identify and demonstrate how these markers change and/or are different with respect to the cause of death and prophylactic treatment but also provide important insights into the potential mechanisms at play during these disease states.

Major comments:

1. It would be useful to include a figure summarising the patient cohort, how they were stratified especially with respect to the stratification of the sub-study (enhanced vs standard; was this only in the group that died or all patients?) and type of analyses performed.

Response: We have now included a CONSORT diagram (Suppl Fig 1) which shows how participants were selected into the substudy and how many were in each randomized group.

2. Figures are of low resolution and in some, almost impossible to make out the labels particularly for the Supplementary figures.

Response: We apologise for the figure quality which we have improved in the resubmission.

3. There are no figure legends (at least in this reviewer's copy) for any of the main figures or supplementary figures.

Response: These have been included in the resubmission.

4. In all figures, please include individual patient data points for transparency of spread and variation.

Response: The revised figures have been reworked to address this point.

5. From Table 2, it is claimed that IL-6 was associated with increased all-cause mortality however p-values from these comparisons was 0.05 and therefore not significant.

Response: In this case the p-value for IL-6 was just below 0.05 (0.0498); we have updated the table with the unrounded value to show that it is below the significance threshold. As all these analyses are exploratory and post-hoc, we do feel that it is important to avoid rigidity around arbitrary thresholds

to draw our inferences. Collectively, we interpret these findings as evidence of an association between IL-6 and mortality, as has been reported in multiple previous studies.

6. In the cluster analysis (Fig 2), it is not immediately obvious what mediators are significant amongst groups. While the titles on each graph indicate ($p < 0.001$), without a figure legend and statistic bars, it is difficult to ascertain what is different.

Response: The P values show that there is strong evidence of difference between groups for each biomarker. Formally to identify pairwise differences between each of the 4 subgroups identified by the pre-specified Calinski-Harabasz stopping rule we would need to follow the closed testing procedure of Marcus *et al* (Biometrika 1976, <https://www.jstor.org/stable/2335748>), whereby for any biomarker with overall difference between the 4 subgroups, we tested for difference between each of the three subsets of 3 subgroups, then tested pairwise differences between each pair within triplets with evidence of difference. The challenge is the sheer number of tests this leads to for each of the 23 biomarkers considered and difficulties in presenting information to the reader. As these analyses are designed only to identify whether there is any evidence for different underlying “phenotypes” in the baseline biomarker data, our preference would be to keep the overall p-values only, but would of course be happy to reconsider if the Editors preferred.

7. With regard to analysing markers in those that received the enhanced vs standard treatment, if it has not been done so (see comment 1), it would be revealing to segregate these groups further into those that survived and those that died given the differences in mortality rates between these treatments.

Responses: The number of participants in the longitudinal analysis does decline over time due to deaths, which were higher in the standard prophylaxis group. The majority of deaths occurred early (now added to Results), hence it is not possible to analyse differences in the changes over time between those who died and survivors.

Minor comments:

- In Table 1, the control group varies between variables examined, i.e for age, sex, CD4 count, BMI variables, this is assumed to be those who survived but for the enhance prophylaxis variable, this is presumably the standard-of-care treatment. It would greatly help interpretation of results if these control groups are clearly defined. Perhaps by asterisks/symbols within the table and the compared groups indicated at the bottom.

Response: In Table 1, cases were participants who died, and controls were participants who survived. We have amended the title, changed the headers, and changed the footnote of this table to clarify this, reflecting lack of clarity about the design highlighted by reviewers 1 and 2 above. The P values compare these two groups, showing that those who died versus survived were older, with lower CD4 counts and lower BMI. We also show how deaths were split by trial arm; as per the original trial findings, mortality was higher in the standard prophylaxis group compared to the enhanced prophylaxis group.

- Include abbreviation for myeloperoxidase (MPO) in main text.

Response: This is now added

- In text referring to supplementary Figure 2, from the cluster analysis, it is indicated that IL-7 is low in Group 2 however there is no IL-7 in the figure.

Response: Thank you for spotting this error; we restricted the number of panels shown in the figure, and it makes sense to highlight biomarkers that are shown in Figure 2. We have therefore amended this text to read: “Group 1 (n=264) had relatively low levels of RANTES, IP-10, stromal cell-derived factor 1 (SDF1a), and growth-regulated alpha protein (GROA).”

•On the graph, the levels for RANTES for Group 3 appear to be to beyond the range of the y-axis. Or are these levels above the detection limit?

Response: For analysis, values above the limit of detection were set at that limit, as stated in the statistical analysis section. All values in group 3 were at this limit. We have clarified this in the Figure footnote.

•“Mortality was lower in group 4 (19%) compared to groups 1, 2 and 3 (32%, 32% and 37%, respectively; $P=0.007$). Causes of death were broadly similar across all 4 groups.” No graphical data for this is presented in the manuscript and would be useful to show as there is space in the figure for it.

Response: We have added this plot to Figure 3.

REVIEWER COMMENTS

Reviewer #2 (Remarks to the Author):

The authors have adequately addressed all my comments, except the one on multiple comparison adjustment. I'm not convinced that it is not needed in the nature of post-hoc analyses. Will still request the authors to consider appropriate multiple comparison adjustment that account for correlations amongst the factors.

Reviewer #3 (Remarks to the Author):

The authors improved considerably manuscript and addressed my queries.

Authors' response to reviewers

Reviewer 2

The authors have adequately addressed all my comments, except the one on multiple comparison adjustment. I'm not convinced that it is not needed in the nature of post-hoc analyses. Will still request the authors to consider appropriate multiple comparison adjustment that account for correlations amongst the factors.

Response: We accept the reviewer's concern, but we are not aware of a global method that could make multiple comparison adjustment across the many different analyses performed and reported in the manuscript, accounting for the correlation between these different analyses and estimators. If the reviewer has a specific method in mind, we would be very happy to consider it.

However, to take the reviewer's point into account, we have considered each analysis presented in the Results in turn, specifically:

- Table 1 shows univariable associations between mortality and baseline characteristics, including 44 biomarkers and 8 patient characteristics. We do not consider it appropriate to apply any p-value adjustment to the patient characteristics since these are well recognised to be associated with mortality in HIV-infected individuals starting treatment outside of this study and therefore should be adjusted for regardless of significance. For the 44 biomarkers, a naïve Bonferroni adjustment would be a p-value of $0.05/44=0.00114$. A more efficient Benjamini-Hochberg approach would be to order the 44 p-values and consider $0.05/44$, $2 \times 0.05/44$, $3 \times 0.05/44$ etc (noting that this still does not address correlation between these tests). Eight tests have $p < 0.001$, meaning the next p-value thresholds to compare to would be $9 \times 0.05/44=0.0102$, $10 \times 0.05/44=0.0114$ and $11 \times 0.05/44=0.0125$. We have therefore indicated with a new symbol the 10 tests which meet these criteria and added a footnote to Table 1 stating "44 biomarkers tested: naïve Bonferroni significance threshold = $0.05/44=0.00114$; symbol indicates tests passing an ordered Benjamini-Hochberg (BH) threshold". We have removed from the main text the biomarker (IL-6) that did not meet the BH threshold.
- Table 2: this table shows the results of a backwards elimination with an exit p-value of 0.1 for 35 biomarkers (excluding the stool biomarkers and those where >40% of values were outside limits of detection). We do not consider it would be appropriate to change the exit p-value now. We have therefore indicated with a new symbol the biomarker p-values which meet the BH threshold in the table and added a footnote for this symbol stating "Biomarkers identified using backwards elimination from 35 non-stool biomarkers with exit $p=0.1$; symbol indicates those meeting an ordered BH threshold". Four are ≤ 0.001 , vs Bonferroni 0.0014, so next (fifth) would be 0.00714 [pass] 0.00857 [fail] so everything except IL-2 and IL-6 has the symbol applied. As we pre-specified the $p=0.1$ exit criteria in this backwards elimination as an exploratory analysis, as described in the Methods and now added to the Results, at least in part to ensure that we adjusted for confounding between biomarkers (noting that associations with IL-6 have been identified in other studies, and hence there is some external support for this finding from the multivariable model after backwards elimination), we have kept the IL-6 and IL-2 associations in the main text, but moved them to a separate sentence and noted that the evidence was weaker for these two biomarkers.
- Figure 1: We have applied the same approach to the cause-specific models in Figure 1, noting that with fewer events (deaths from specific causes), power is intrinsically lower and therefore significance does not necessarily imply lack of effect, but may simply reflect lower

power. In particular Figure 1 makes it clear that many effect sizes are similar in cause-specific models as for all-cause mortality (eg for CRP for TB and severe bacterial infection deaths) and therefore we have not amended the main Results text other than noting the power limitation.

- Best-fitting subsets regression: this approach considers whether the models selected by backwards elimination are well-supported across the population of all possible models that could have been fitted. Multiple testing is therefore not relevant.
- Figure 2: We have added the number of biomarkers that had strong evidence ($p < 0.001$) for differences between the four clusters identified from hierarchical clustering
- Table 3: this table shows the results of interaction tests, aiming to assess whether there is any evidence that change from baseline in 44 biomarkers differed according to randomisation to standard vs enhanced prophylaxis. The challenge here is again that interaction tests are known to have low power, because they are testing for a difference in the effect of a factor (here randomised group) on change from baseline, i.e. that the change from baseline differs by randomised group. No interaction test meets the naïve Bonferroni threshold, and hence none meets the BH threshold, by definition. We have therefore added a symbol to the p-value header with a footnote stating this “No test meets the ordered BH threshold” and added this point to the main text.

Reviewer 3

The authors improved considerably manuscript and addressed my queries.

Response: Thank you.

REVIEWERS' COMMENTS

Reviewer #2 (Remarks to the Author):

Thank you for addressing the additional comments. I agree that indicating which markers would meet the BH-adjusted p-value criterion is sufficient.